# Study on Subjective Evaluation of Acoustic Environment in Urban Open Space Based on “Effective Characteristics”

**DOI:** 10.3390/ijerph19159231

**Published:** 2022-07-28

**Authors:** Xiaodan Hong, Weichen Zhang, Yiping Chu, Wenying Zhu

**Affiliations:** 1Shanghai Academy of Environmental Sciences, Shanghai 200233, China; zhangwc@saes.sh.cn (W.Z.); choiychu@163.com (Y.C.); 2Shanghai Engineering Research Center of Urban Environmental Noise Control, Shanghai 200233, China

**Keywords:** acoustic environment quality evaluation, urban open space, subjective satisfaction, the satisfaction evaluation model of acoustic environment, “effective characteristics”, multivariable linear regression algorithm

## Abstract

With the continuous expansion of urban scale with dense population and traffic and the gradual improvement of residents’ requirements for environmental quality, the traditional evaluation method relying on acoustic energy is not enough to reflect the feelings of urban crowds about acoustic environment quality. The acoustic environment quality evaluation method based on human subjective perception has gradually become one of the research focuses in the field of environmental noise control. In recent years, various subjective and objective acoustic characteristic parameters have been introduced into the study of acoustic environment assessment in the global literature. However, the extraction of “effective characteristics” from a large number of physical and psychoacoustic characteristics contained in acoustic signals and the creation of a scientific and efficient subjective evaluation model have always been key technical problems in the field of acoustic environment evaluation. Based on subjective human perceptions, the overall acoustic environment quality evaluation of urban open spaces is studied in this paper. Based on the “effective characteristic” parameters and the subjective characteristic proposed in the previous research, including equivalent continuous A-weighted sound pressure level (*L_A_*), the difference between median noise and ambient background noise (*L*_50_ − *L*_90_), Sharpness (*Sh*), as well as satisfaction (*Sat*), the multivariable linear regression algorithm is used to further study the intrinsic correlation between the proposed “effective characteristics” and subjective perception. Then, a satisfaction evaluation model of the acoustic environment based on “effective characteristics” is built in this paper. Furthermore, the soundwalk evaluation experiment and the MATLAB numerical simulation experiment are carried out, which verify that the prediction accuracy of the proposed model is more than 92%, the consistency of satisfaction level is more than 88%, as well as the changes in the values of *Sh* and *L*_50_ − *L*_90_ have a significant impact on the satisfaction prediction of the proposed model. It shows that the proposed “effective characteristics” more comprehensively describe the quality level of the regional acoustic environment in urban open space compared with a single *L_A_* index, and the proposed acoustic environment satisfaction evaluation model based on “effective characteristics” has significant accuracy superiority and regional applicability.

## 1. Introduction

In a report about the impact of noise on health, i.e., the disease burden caused by noise pollution, published by the cooperative research center of the World Health Organization and European Union, it was pointed out that the disease burden caused by noise pollution is second only to air pollution [1,2]. With the continuous acceleration of China’s urbanization level, the urban scale of densely populated cities represented by Shanghai, Beijing, Guangzhou and Shenzhen has been continuously expanding and the population density is increasing in recent years, which has brought huge traffic flow and dense buildings. Throughout the layout of large- and medium-sized cities, residential areas are generally distributed along both sides of urban primary and secondary traffic arteries. Urban open spaces (including residential areas, urban public green spaces and other areas for residents’ daily leisure and activities) are inevitably affected by all kinds of traffic noise. Urban noise pollution has become an important environmental factor affecting quality of life and human health [3,4].

Traditionally, physical indicators, such as equivalent continuous A-weighted sound pressure level (*L_A_*) are used to quantitatively evaluate noise. However, the main impact of noise on humans is to cause subjective feelings, such as annoyance [5,6]. From the perspective of auditory perception, the ultimate goal of noise control is to reduce the annoyance caused by noise. In recent years, scholars have begun to pay more attention to “people-oriented,” explore the evaluation method based on subjective perception, and try to apply it to the field of urban acoustic environment quality control [7,8,9,10].

Research on the subjective evaluation of the acoustic environment began in the 1970s. Schultz obtained the exposure response relationship of noise by data fitting between the annoyance ratio of residents and the day and night equivalent sound pressure level Ldn in many European and American cities [11]. Since then, studies on the relationship between sound pressure level characteristics and annoyance responses have emerged at home and abroad. However, these studies have always focused on the impact of a single *L_A_* on subjective annoyance [12,13,14,15,16,17,18]. At the end of the 20th century, A. Kjellberg, K. Persson, J.S. Bradley and other researchers found that other components of sound have a certain impact on human subjective feelings [19,20,21]. A single study on the impact of *L_A_* on subjective perceptions makes it difficult to fully reflect real human feelings about the acoustic environment. At the beginning of the 21st century, scholars have considered various (physical and psychological) characteristics other than sound pressure level to study the subjective evaluation of the acoustic environment (in a specific space), and some research results have been achieved [7,22,23,24]. In 2009, Yu and Kang of the University of Sheffield built the sound pressure level evaluation model and sound comfort evaluation model of British city squares based on sound pressure levels [25,26]. Since then, subjective evaluation models of acoustic environments based on physical and psychoacoustic characteristics have gradually become research hotspots in the acoustic subjective evaluation field. Aiming at the independent sound sources in residential areas, Lu discussed the acoustic comfort evaluation model based on the characteristic parameters of sound pressure level, loudness and clarity [27]. In recent years, considering acoustic objective parameters such as sound pressure level, loudness, roughness, sharpness, tone and shaking degree, the Kang J team has explored sound comfort evaluation models for single sound sources, including running water sound, bird sound and car sound [28]. On this basis, Yu and Xu further built a sound source evaluation model [29].

Throughout the research status at home and abroad, most of the current research on subjective evaluation of the urban acoustic environment is still focused on a single or similar sound source (such as running water sound, car sound, bird singing, etc.) [28,29,30], as well as a single urban space type, such as a specific urban square, park, residential area, street, commercial street, etc. [25,26,27,30]. These small-scale studies are difficult to apply to evaluate the overall urban acoustic environment with complex and mixed sound sources. Moreover, on the one hand, existing studies often separate the sound pressure level from other acoustic characteristics and divide the subjective acoustic evaluation into sound level evaluation and sound source evaluation [25,26,28,29,31], ignoring the interaction between the acoustic characteristics and their compound influence on human subjective perception. On the other hand, existing studies often use various acoustic characteristics with significant correlations to build the subjective evaluation model at the same time [28,29], which makes the model with a large amount of redundant information, makes it difficult to obtain the effective characteristics for subjective evaluation of the acoustic environment, and increases the complexity of the evaluation model. Therefore, the extraction of “effective characteristics” and the creation of a scientific and efficient subjective evaluation model that is applicable to the overall urban acoustic environment evaluation have always been the key technical problems to be solved.

In this paper, the “effective characteristics” and the subjective satisfaction model for evaluating the overall quality of the urban acoustic environment are studied quantitatively. Based on effective physical and psychoacoustic characteristic parameters and subjective parameters obtained in previous studies [32,33,34,35,36,37], adopting the multivariate linear regression modeling method, the paper builds a subjective satisfaction evaluation model of the acoustic environment based on “effective characteristics”. The soundwalk evaluation comparison experiment and MATLAB numerical simulation experiment are employed to verify the applicability and superiority of the proposed satisfaction model. The research expands the spatial scale and sound source complexity of the subjective evaluation and also provides a technical basis for establishing the scientific and efficient evaluation indexes and evaluation methods of urban acoustic environment quality.

The rest of the paper is organized as follows. In Section 2, firstly, the “effective characteristic” parameters for acoustic environment evaluation are explained and a sample set of “effective characteristics” − “satisfaction” is constructed. Then, a satisfaction evaluation model based on “effective characteristics” is proposed in this section. Verification of the performance of the proposed satisfaction evaluation model is performed in Section 3 by employing a soundwalk evaluation experiment and a MATLAB numerical simulation test. In Section 4, the conclusions are presented.

## 2. Materials and Methods

### 2.1. Case Study

In this paper, a subjective evaluation of the overall acoustic environment of urban open spaces is conducted. In the early stages of the study, focusing on class 2 sound functional areas [38] with concentrated residents’ activities in the central urban area and outside the outer ring road of Shanghai, about 50 typical residential areas and open green spaces were selected successively, with a total of 63 points for on-site measurement. Then, 63 groups of acoustic signals of typical open areas representing the features of the overall acoustic environment of Shanghai were collected. The range of LA is 45~63 dB(A), as shown in Figure 1.

### 2.2. Methodology 

In the early stage, the team constructed the “characteristics” − “satisfaction” sample set through the characteristics analysis of the overall urban acoustic environment with complex and mixed sound sources in Shanghai open spaces and the subjective evaluation experiment in the laboratory. Based on the sample set, a dual correlation coefficient evaluation method is proposed to study the correlations between 17 objective and subjective characteristics, and then the “effective characteristic” parameters applicable to the subjective evaluation of the overall urban acoustic environment are proposed [32].

Based on previous achievements [32,33,34,35,36,37], aiming at the overall urban open space, the potential correlation between the “effective characteristics” and subjective “satisfaction” is quantitatively studied in this paper by adopting the multivariable linear regression modeling method. Then, a satisfaction evaluation model of the acoustic environment based on the “effective characteristics” is built. The coefficients of the evaluation model are optimized using the least square method based on the MATLAB platform. Finally, the soundwalk evaluation comparison experiment and MATLAB numerical simulation experiment are employed to verify the prediction accuracy and real scene applicability of the proposed subjective satisfaction model of the urban acoustic environment. A flowchart of the methodology is shown in Figure 2.

### 2.3. Construct of the “Effective Characteristics” − “Satisfaction” Sample Set

Based on the collected 63 groups of typical acoustic signals, on the one hand, Artemis software (V12. 02. 000, from HEAD acoustics GmbH, Herzogenrath, Germany) is used to analyze and extract 16 kinds of acoustic objective characteristics. Then, 63 groups of objective characteristic parameters of the acoustic environment are obtained as ${\left\{F_{i}\right\}}_{i=1}^{63}$, where, the expression of the $i\left(=1,\cdots ,63\right)$-th characteristic vector $F_{i}$ is:(1)$$F_{i}=\left(f_{i1},\cdots ,f_{ij},\cdots f_{i16}\right)={\left(L_{5},L_{10},L_{50},L_{90},L_{1}{}_{0}-L_{90},L_{50}-L_{90},L_{A},L_{C},L_{C}-L_{A},D_{250},D_{315},D_{50}{}_{0},Lou,R,Flu,Sh\right)}_{i}$$

The 16 kinds of objective characteristics include physical parameters as *L*_5_, *L*_10_, *L*_50_, *L*_90_, *L*_10_ − *L*_90_, *L*_50_ − *L*_90_, *L_C_*, *L_A_*, *L_C_* − *L_A_*, *D*_250_*/D*_315_*/D*_500_ (energy difference between octave band of 250 Hz/315 Hz/500 Hz and below and octave band above 250 Hz/315 Hz/500 Hz), as well as psychoacoustic parameters as *Lou *(Loudness), *R* (Roughness), *Flu* (Fluctuation), *Sh* (Sharpness). They are sorted out from various acoustic factors in the global literature and standard guidelines [19,20,21,28,29,34,39,40]. On the other hand, the seven-level semantic subdivision method is used to carry out the laboratory subjective evaluation experiment on 63 groups of acoustic signals, and 63 groups of subjective “satisfaction” samples $SAT={\left\{Sat_{i}\right\}}_{i=1}^{63}$ are obtained. Finally, the “characteristics” − “satisfaction” sample set ${\left\{\left(F_{i},Sat_{i}\right)\right\}}_{i=1}^{63}$ of the overall acoustic environment of open space in Shanghai was established [32]. The subjective evaluation laboratory and experimental instruments are shown in Figure 3. Figure 4 lists the subjective evaluation experiment based on the seven-level semantic subdivision method. The degree descriptors are divided into very dissatisfied, relatively dissatisfied, slightly dissatisfied, neutral, slightly satisfied, relatively satisfied and very satisfied, which correspond to the satisfaction values of 1~7, respectively.

Then, based on the established “characteristics” − “satisfaction” sample set ${\left\{\left(F_{i},Sat_{i}\right)\right\}}_{i=1}^{63}$, the dual correlation coefficient evaluation method is proposed, which uses Pearson’s correlation coefficient r to quantitatively evaluate the autocorrelation coefficients $r_{jk}\left(j,k=1,\cdots 16\right)$ between 16 objective characteristic parameters $F={\left\{f_{j}\right\}}_{j=1}^{16}$ and the correlation coefficients $r_{j}\left(Sat\right)\left(j=1,\cdots 16\right)$ of 16 objective characteristics with the subjective *Sat*. The Equations of $r_{jk}$ and $r_{j}\left(Sat\right)$ can be described as follows:(2)$$r_{j}\left(Sat\right)=\sum_{i=1}^{63}(f_{ij}-{\bar{f}}_{j})\left(Sat_{i}-S\bar{a}t\right)/\sqrt{\sum_{i=1}^{63}{\left(f_{ij}-{\bar{f}}_{j}\right)}^{2}\sum_{i=1}^{63}{\left(Sat_{i}-S\bar{a}t\right)}^{2}}$$
(3)$$r_{jk}=\sum_{i=1}^{63}(f_{ij}-{\bar{f}}_{j})\left(f_{ik}-{\bar{f}}_{k}\right)/\sqrt{\sum_{i=1}^{63}{\left(f_{ij}-{\bar{f}}_{j}\right)}^{2}\sum_{i=1}^{63}{\left(f_{ik}-{\bar{f}}_{k}\right)}^{2}}$$
where $j,k\left(=1,\cdots 16\right)$ are the number of objective characteristic parameters, $r_{j}\left(Sat\right)$ represents the correlation coefficient between the *j-*th characteristic and *Sat*, and $r_{jk}$ represents the correlation coefficient between the *i-*th characteristic parameter and the *j-*th characteristic parameter. Therefore, using the MATLAB platform, the correlation coefficient matrices $R_{1\times 16}$ and $R_{16}$ are calculated:(4)$$R_{1\times 16}=\left[r_{1}\left(Sat\right),\cdots r_{j}\left(Sat\right),\cdots r_{16}\left(Sat\right)\right]$$
(5)$$R_{16}=\left[\begin{array}{cccc} r_{1,1} & r_{1,2} & \cdots & r_{1,16}\\ r_{2,1} & r_{2,2} & \cdots & r_{2,16}\\ \vdots & \vdots & \ddots & \vdots \\ r_{16,1} & r_{16,2} & \cdots & k_{16,16} \end{array}\right]$$

Finally, the “effective characteristic” parameters applicable to subjective evaluation of the overall urban acoustic environment are proposed, including equivalent continuous A-weighted sound pressure level (*L_A_*), the difference between median noise and ambient background noise (*L*_50_ − *L*_90_) and Sharpness (*Sh*). The correlation studies show that *L_A_* is the decisive factor affecting subjective satisfaction, and its negative correlation with *Sat* reaches 88%, the negative correlation between *L*_50_ − *L*_90_ and *Sat* is 25%, and the positive correlation between *Sh* and *Sat* is 33%. Table 1 and Table 2 exhibit the correlation coefficients of 16 objective characteristics with subjective parameter *Sat*, as well as the autocorrelation coefficients between 16 objective characteristics [32].

In order to quantitatively study the internal relationship between the proposed “effective characteristic” parameters and human subjective “satisfaction,” and build a scientific and efficient subjective evaluation model of the acoustic environment, this paper reduces the dimensions of 63 groups of original “characteristics” − “satisfaction” experimental samples ${\left\{\left(F_{i},Sat_{i}\right)\right\}}_{i=1}^{63}$, and then constructs 63 groups of “effective characteristics” − “satisfaction” samples ${\left\{\left(C_{i},Sat_{i}\right)\right\}}_{i=1}^{63}$. Where $C={\left\{C_{i}\right\}}_{i=1}^{63}={\left\{{\left(L_{50}-L_{90},L_{A},Sh\right)}_{i}\right\}}_{i=1}^{63}$ represents 63 “effective characteristics” samples. $SAT={\left\{Sat_{i}\right\}}_{i=1}^{63}$ consists of 63 groups of satisfaction values. Based on the 63 groups of effective characteristic vectors $\left\{C_{1},\cdots ,C_{i},\cdots C_{63}\right\}$, combined with the variation range of characteristic parameters of acoustic environment samples in the overall open area of Shanghai, the numerical interval of “effective characteristic” parameters is evaluated as $C_{i}\in \left[C_{min},C_{max}\right]$, where $C_{min}=\left(0.3,45,1\right)$ and $C_{max}=\left(6,65,2.5\right)$.

### 2.4. Building an Acoustic Environment Satisfaction Evaluation Model Based on “Effective Characteristics”

Based on the experimental sample set ${\left\{\left(C_{i},Sat_{i}\right)\right\}}_{i=1}^{63}$ of “effective characteristics” − “satisfaction”, this section adopts the multivariable linear regression modeling method to build the acoustic environment satisfaction evaluation model based on “effective characteristics”. The coefficients of the model are regressed and optimized using the least square method and its residual optimization.

#### 2.4.1. Multivariable Linear Regression Modeling Based on the Least Square Algorithm

Employing the multivariable linear regression algorithm to build an acoustic environment satisfaction model is as follows: (6)$$Sat=\theta_{0}+\theta_{1}*\left(L_{50}-L_{90}\right)+\theta_{2}*L_{A}+\theta_{3}*Sh+e$$
where, $\theta_{0}$ is the constant term, $\theta_{1},\theta_{2},\theta_{3}$ are the partial regression coefficients, and e represents the random error. Based on 63 groups of “effective characteristics” − “satisfaction” experimental samples ${\left\{{\left(L_{50}-L_{90},L_{A},Sh,Sat\right)}_{i}\right\}}_{i=1}^{63}$, using the linear regression Equation (6), the matrix expression of the satisfaction evaluation model can be written as follows:(7)$$Sat_{\theta }\left(X\right)=X\theta$$
where, satisfactions $Sat_{\theta }\left(X\right)$ is the 63×1 dimensional output vector, θ is a 4×1 parameter vector to be estimated, and X is the 63×4 input matrix. The specific expressions are as follows:(8)$$X={\left[\begin{array}{cccc} 1 & {\left(L_{50}-L_{90}\right)}_{1} & {\left(L_{A}\right)}_{1} & Sh_{1}\\ \vdots & \vdots & \vdots & \vdots \\ 1 & {\left(L_{50}-L_{90}\right)}_{63} & {\left(L_{A}\right)}_{63} & Sh_{63} \end{array}\right]}_{63\times 4}$$
(9)$$\theta ={\left[\begin{array}{c} \theta_{0}\\ \theta_{1}\\ \theta_{2}\\ \theta_{3} \end{array}\right]}_{4\times 1}$$

The parameter θ in Equation (6) is estimated by the least square method. The least square method was discovered by A. M. Legendre in the 19th century. Its main idea is to calculate the unknown parameters to minimize the square sum of the differences e between the prediction values and the experimental values (i.e., random error or residual). Therefore, the principle of the least square method is to obtain the fitting function model when minimizing the loss function E, which can be calculated by the following Equation:(10)$$E=\sum_{i=1}^{63}e_{i}^{2}=\sum_{i=1}^{63}{\left(Sat_{i}-S\hat{a}t_{i}\right)}^{2}$$
where the prediction values $\left\{S\hat{a}t_{i}|i=1,\cdots ,63\right\}$ represent the fitting function values of the regression model, $\left\{Sat_{1},\cdots ,Sat_{63}\right\}$ are 63 satisfaction experimental values. Our goal is to minimize the objective function E (i.e., loss function) and then obtain the fitting function model. The following are the matrix-solving processes.

The loss function is defined as follows:(11)$$J\left(\theta \right)=1/2{\left(X\theta -SAT\right)}^{T}\left(X\theta -SAT\right)$$
where $SAT={\left(Sat_{1},\cdots ,Sat_{63}\right)}^{T}$ is a 63-dimensional output vector composed of satisfaction experimental values. For the convenience of calculation, the coefficient 1/2 is selected in Equation (11) so that the coefficient after derivation is equal to 1. According to the principle of the least square method, the derivative of the loss function $J\left(\theta \right)$ with respect to vector θ is assumed to be equal to 0, and then Equation (12) is obtained:(12)$$\partial J\left(\theta \right)/\partial \theta =X^{T}\left(X\theta -SAT\right)=0$$

Considering the chain rule of matrix derivation and two equations of matrix derivation, which are described as follows:

Equation (1): $\partial \left(X^{T}X\right)/\partial X=2X$;

Equation (2): $\nabla_{X}f\left(AX+B\right)=A^{T}\nabla_{Y}f$, Y=AX+B, and $f\left(Y\right)$ is a scalar.

Finally, after deriving the above Equation (12), Equation (13) can be obtained:(13)$$X^{T}X\theta =X*SAT$$

Multiply ${\left(X^{T}X\right)}^{-1}$ on both sides of Equation (13), at the same time, the parameter vector $\theta_{4\times 1}$ is calculated with the following Equation:(14)$$\theta ={\left(X^{T}X\right)}^{-1}X^{T}*SAT$$

#### 2.4.2. Model Regression and Optimization Based on MATLAB Platform

Considering 63 groups of experimental samples ${\left\{\left(C_{i},Sat_{i}\right)\right\}}_{i=1}^{63}$, based on the MATLAB platform, this section uses the least square method and its residual optimization to estimate parameters θ of the built multivariable linear regression model in Equation (7). A time series residual diagram (shown in Figure 5) is used for residual analysis. After eliminating singular points six times, 51 groups of training samples remained. Finally, the acoustic environment satisfaction evaluation model based on “effective characteristics” is obtained:(15)$$S\hat{a}t=16.93-0.017*\left(L_{50}-L_{90}\right)-0.28*L_{A}+1.3*Sh$$

The values of the estimated parameters are $\theta ={\left(16.93-0.017,-0.28,1.3\right)}^{T}$. The specific regression results are shown in Table 3, which shows that the correlation coefficient $R^{2}(=0.9347)$ of the multivariable linear regression model is close to 1, and $F_{1-\alpha }\left(3,47\right)=2.8024<F(=224.4254)$, $p_{F}<0.0001$, which demonstrate that the effectiveness of the proposed satisfaction evaluation model is significant.

Further, in order to verify the effectiveness and prediction accuracy of the regression model after six times of residual optimization, the error variances *S*^2^, the Root Mean Square Errors (*RMSE*), and the correlation coefficients $R_{\left(Sat,S\hat{a}t\right)}$ between the predictions and the experimental values of subjective satisfaction are considered the model performance criteria. *RMSE* and $R_{\left(Sat,S\hat{a}t\right)}$ are calculated using the following Equations [32,41,42]:(16)$$RMSE=\sqrt{\sum_{i=1}^{n}{\left(Sat_{i}-S\hat{a}t_{i}\right)}^{2}/n}$$
(17)$$R_{\left(Sat,S\hat{a}t\right)}=\sum_{i=1}^{63}(Sat_{i}-S\bar{a}t)\left(S\hat{a}t_{i}-S\bar{\hat{a}}t\right)/\sqrt{\sum_{i=1}^{63}{\left(Sat_{i}-S\bar{a}t\right)}^{2}\sum_{i=1}^{63}{\left(S\hat{a}t_{i}-S\bar{\hat{a}}t\right)}^{2}}$$
where $i\left(=1,\cdots ,n\right)$ is the test data time index, n is the number of the test samples, $Sat_{i},S\hat{a}t_{i}$ are the experimental value and the prediction value at the i-th sample, respectively.

The error variances *S*^2^, *RMSE* values and correlation coefficients $R_{\left(Sat,S\hat{a}t\right)}$ of the initial regression model and the regression models after six optimizations are listed in Table 4. Obviously, after six times of residual optimization, the *S*^2^ and *RMSE* values of the regression model are significantly reduced, and $R_{\left(Sat,S\hat{a}t\right)}$ between the experimental values and the prediction values is increased to 96.7%, which shows that the optimized satisfaction evaluation model of the acoustic environment has obvious accuracy superiority.

## 3. Real Scene Application Verification of Acoustic Environment Satisfaction Evaluation Model

### 3.1. Soundwalk Evaluation Experiment

The soundwalk evaluation method is a commonly used experimental method for sound subjective research and sound scene research [43,44]. The soundwalk evaluation method requires experimenters to evaluate the sound environment at the designated place in the urban green space and the hinterland of the residential area, and the acoustic signals are collected simultaneously. According to the international standard of soundscape ISO/TS 12913-2:2018 [45], to prevent mutual interference, the number of soundwalk experimenters should be about 5. The evaluation points shall be set in advance according to the site conditions. It is necessary to select the points that can represent the characteristics of the acoustic environment in the area, ensure the consistency of the environment evaluated by the experimenters and avoid the interference of sudden noise events. Compared with the laboratory evaluation method, the soundwalk evaluation method can better reflect the intuitive feelings of the crowd in the current acoustic environment [43,44,46].

In this paper, three representative open spaces, including the viaduct, ground road and rail transit line, are randomly selected for the soundwalk experiment on site to obtain the actual evaluation values of the crowd. The acoustic environment signals of the points in the three spaces are collected synchronously; then, the corresponding prediction values of satisfaction are calculated by using the satisfaction evaluation model proposed in this paper. At the same time, the *L_A_*-*Sat* evaluation model (i.e., the linear fitting function of subjective satisfaction *Sat* on a single characteristic *L_A_*) is used to predict the acoustic environment satisfaction of these points. Then, the superiority of the proposed subjective evaluation model based on “effective characteristics” is verified in this section. On the one hand, by comparing the deviation degree and satisfaction level consistency between the prediction values of the proposed model and the soundwalk evaluation values, the prediction accuracy and field applicability of the proposed satisfaction evaluation model are evaluated. The superiority of the proposed satisfaction model is further verified by comparing the prediction accuracy with the *L_A_*-*Sat* evaluation model. On the other hand, the MATLAB numerical simulation test is used to evaluate the impact of characteristics *L*_50_ − *L*_90_ and *Sh* on subjective satisfaction *Sat* when the value of *L_A_* remains unchanged, so as to verify the effectiveness of “effective characteristics” for the evaluation of urban acoustic environment quality.

The design of the experimental points and experimental schemes of the field soundwalk experiment are shown in Table 5. The scenes of the soundwalk evaluation experiment are shown in Figure 6, including soundwalk evaluation, scene recording, and sound signal collection. A total of 6 experimenters (including 4 men and 2 women) were randomly selected from the experimental personnel who participated in the subjective evaluation in the laboratory in the early stage. They evaluated the degree of satisfaction with the acoustic environment in different periods according to the gestures of the soundwalk conductor. The seven-level semantic subdivision method is still used. Figure 7 shows the subjective evaluation list of the soundwalk experiment.

### 3.2. Analysis of Prediction Accuracy and Superiority of the Model in Real Scenes

The correlation analysis is carried out on the soundwalk evaluation results of 6 experimenters. The average value of the correlation coefficient is 0.77, and the lowest value is 0.67. That is, the evaluation results of 6 experimenters are basically the same, which are all valid data.

#### 3.2.1. Accuracy and Regional Applicability Verification of the *Effective Characteristics-Sat* Model

On the one hand, to compare the prediction accuracy between the proposed *Effective Characteristics-Sat* model and *L_A_-Sat* evaluation model, the performance index Mean Absolute Percentage Error (*MAPE*) is used to evaluate the deviation degree between the prediction values of the models and the soundwalk evaluation values, which can be calculated as Equation (18). On the other hand, in order to show the consistency between the prediction results of the proposed model and the evaluation results of soundwalk more simply and intuitively, and reduce the complexity of the evaluations of the model (improve the practicability of the proposed model for the evaluation of urban regional acoustic environment quality), it is considered to simplify the satisfaction evaluation results of levels 1~7 into three level intervals of “dissatisfaction,” “neutral “ and “satisfaction,” so as to verify the consistency of subjective satisfaction levels. Specifically, by ranking the 63 groups of satisfaction evaluation values of laboratory subjective experiments from low to large and counting the evaluation results of the seven-level semantic subdivision method, it is found that the “neutral” evaluations of experimenters are basically concentrated near the score of 4. Therefore, 3.32~4.68 is defined as the “neutrality” interval, which corresponds to the number of experimenters evaluating “neutral” (i.e., score 4) as close to half. Then, the satisfaction evaluation values of levels 1~7 can be simply divided into “neutrality” with an interval of 3.32 < *Sat* < 4.68, “dissatisfaction” interval of *Sat* ≤ 3.32 and “satisfaction” interval of *Sat* ≥ 4.68 [33].
(18)$$MAPE=\frac{1}{m}\sum_{i=1}^{m}\left|\left(S\hat{a}t_{i}-Sat_{i}\left(act\right)\right)/Sat_{i}\left(act\right)\right|\times 100\%$$
where *m* represents the number of soundwalk samples, $S\hat{a}t_{i}$ and $Sat_{i}\left(act\right)$ represent the satisfaction prediction value and the soundwalk evaluation value of the *i*-th sample, respectively.

The predictions of the satisfaction evaluation model proposed in this paper and the soundwalk evaluation results for the acoustic environment in three representative open areas are compared in Table 6 and Table 7. Table 6 shows that the *MAPE* value of prediction values of the proposed model and the evaluation values of soundwalk is about 7.66%, and the *MAPE* value of the predictions of the *L_A_-Sat* evaluation model and the soundwalk evaluation values is about 10.06%, indicating that the prediction accuracy of the acoustic environment satisfaction of the soundwalk areas of the proposed evaluation model is 92.34%, which is better than the prediction accuracy of the *L_A_-Sat* model of 89.95%, which verifies the accuracy superiority of the proposed model. As shown in Table 7, after rating the prediction values of the proposed model and the evaluation values of the soundwalk according to 1–3 levels of satisfaction, the consistency of the satisfaction levels is 88.89%. The satisfaction levels of the prediction values of the model and the evaluation values of the soundwalk are basically the same. The two samples with different satisfaction levels listed in Table 7 are for a short one-minute period; the soundwalk evaluations are disturbed by the unsteady part of the short-term sound. After excluding the evaluation samples for a 1-min period, the prediction accuracy of the proposed satisfaction model for the test areas is increased to 93.36% (as shown in Table 6), and the satisfaction levels of the prediction values and the soundwalk evaluation values are completely consistent (as shown in Table 7). In addition, as can be seen from the satisfaction rating results in Table 7, differing from the soundwalk evaluations easily disturbed by short-term sudden noise events, the satisfaction levels of the model prediction values of acoustic environment signals of different time lengths in the same region are consistent. This indicates that the proposed satisfaction evaluation model based on “effective characteristics” has time robustness, and the predictions of the model are less disturbed by the unsteady part of the short-term sound signal. Therefore, based on the verification results in the subsection, the acoustic environment satisfaction evaluation the model based on the “effective features” proposed in this paper has superior prediction accuracy and field applicability. The three-level satisfaction division method can simplify the evaluation results of the proposed model and then improve the practicability of the model. In addition, the prediction accuracy of the model is not limited by the duration of the acoustic signal, which can fully describe the long-term subjective feeling for a regional acoustic environment with a steady-state sound signal for just 10 s. The evaluation method based on model can avoid external interferences that are difficult to eliminate in the traditional soundwalk evaluation.

#### 3.2.2. Superiority Verification of the *Effective Characteristics-Sat* Evaluation Method

To further verify the superiority of the proposed acoustic environment satisfaction evaluation method, in this subsection, the impact of the other two “effective characteristics,” *L*_50_ − *L*_90_ and *Sh*, on the prediction value of subjective satisfaction is evaluated under the condition that the *L_A_* remains unchanged. Because it is difficult to collect a large number of acoustic environment samples with the same *L_A_* in real areas, this paper uses existing experimental samples collected in typical open areas (a total of 81 groups of acoustic signals) to conduct numerical simulation tests with the MATLAB platform. Specifically, considering the *L_A_-Sat* evaluation model depending solely on *L_A_*, the sound level interval corresponding to the subjective evaluation of “dissatisfaction” is obtained as *L**_A_* ≥ 55.76 dB (corresponding to *Sat* ≤ 3.32), and the *L_A_* interval subjectively evaluated as “satisfaction” is *L_A_* ≤ 51.09 dB (corresponding to *Sat* ≥ 4.68). To more simply and intuitively verify the influence of the change of the characteristic parameters *L*_50_ − *L*_90_ and *Sh* on the subjective *Sat* in the proposed model, here, the sound level values *L_A_* of the simulation samples are taken as 55.76 dB(A) and 51.09 dB(A), respectively. In combination with the 81 characteristic values of *L*_50_ − *L*_90_ and *Sh* collected in the previous experiments, two groups of simulation experiment sample sets (each group containing 81 samples) are constructed by using MATLAB. Then, the satisfactions SAT_55.76dB(A)_ and SAT_51.09dB(A)_ of two groups of data samples are predicted using the satisfaction evaluation model based on the “effective characteristics” proposed in this paper.

Figure 8 shows the (3D/2D) change trends of SAT_55.76dB(A)_ and SAT_51.09dB(A)_ on the characteristic parameters *L*_50_ − *L*_90_ and *S**h*. It can be seen from Figure 8 that the values of subjective satisfaction *Sat* are obviously affected by the changes in sharpness *Sh* and *L*_50_ − *L*_90_ when the *L_A_* remains unchanged. As shown in 3D Figure 8a, the higher the value of *Sh* and the lower the value of *L*_50_ − *L*_90_, the higher the value of *Sat*. Moreover, compared with the impact of *L*_50_ − *L*_90_ on *Sat*, the impact of *Sh* on *Sat* is more significant. 2D Figure 8b more intuitively shows the changes in *Sat* of the 81 × 2 samples. In Figure 8b, the first 18 × 2 samples are from the soundwalk experimental samples, the background acoustic environment of these samples is similar, and the values of the characteristics *L*_50_ − *L*_90_ and *Sh* of the acoustic samples differ slightly. Therefore, the change range of the satisfaction values is also small, and they are basically concentrated in the same satisfaction level. Among them, there are four samples (the first and the 16th–18th samples) with obviously smaller values of characteristics *Sh*, so their values of *Sat* are obviously reduced. The last 63 × 2 samples in the figure show significant change trends in the values of *Sat* due to a wide range of collected areas and large changes in the background acoustic environment. In general, as can be seen in Figure 8, the evaluation values *Sat* fluctuate within “dissatisfaction” and “neutrality” depending on the change of the values of characteristics *L*_50_ − *L*_90_ and *Sh* when *L_A_* = 55.76 dB; the evaluation values *Sat* fluctuate within “satisfaction” and “neutrality” with the change of characteristics *L*_50_ − *L*_90_ and *Sh* when *L_A_* = 51.09 dB. Therefore, from the satisfaction prediction results in this subsection, it can be seen that the proposed “effective characteristics” more comprehensively describe humans’ subjective feelings in the regional acoustic environment compared with a single *L_A_* index. Compared with the traditional subjective evaluation method based on sound pressure level, the satisfaction evaluation model based on “effective characteristics” proposed in this paper has higher prediction accuracy for the acoustic environment of various open spaces.

### 3.3. Discussion of Applicability and Superiority

The subjective satisfaction model proposed in this paper is applicable to evaluate the acoustic environment of open space of the urban class 2 sound functional areas (including residential areas, urban public green spaces and other areas for residents’ daily leisure and activities), which are affected by all kinds of traffic noise. Compared with the existing studies on subjective evaluation, which mostly focus on a specific space and single/similar sound source, the proposed model expands the space scale and the complexity of sound sources of subjective evaluation. Compared with the existing research on subjective evaluation models, which either only consider a single sound pressure level or use various significantly related acoustic characteristics to establish an evaluation model at the same time, the proposed subjective evaluation model based on “effective characteristics” improves the evaluation accuracy and eliminates a large number of redundant characteristic parameters. The study simplifies the number of evaluation indicators to the greatest extent and reduces the complexity of the evaluation model. Therefore, the satisfaction evaluation model based on “effective characteristics” proposed in this paper provides technical support for the regional acoustic environment quality evaluation of large- and medium-sized cities and also provides a research direction for the improvement of regional acoustic environment quality standards. In addition, by studying the “modification” and “fine adjustment” of the “effective characteristics” of sounds, the acoustic environment quality of open space will be accurately improved so as to improve people’s quality of life.

## 4. Conclusions

In this paper, the subjective evaluation of the overall acoustic environment of the urban class 2 sound functional open areas (including residential areas, urban public green spaces and other areas for residents’ daily leisure and activities) is studied. In the early stage, the “characteristics” − “satisfaction” sample set was obtained by collecting the acoustic environment signals of various open areas in Shanghai and establishing the subjective evaluation experiment in the laboratory. Then, by studying the dual correlations of 16 kinds of objective characteristic parameters and the subjective parameter “satisfaction,” three “effective characteristic” parameters for subjective evaluation of the acoustic environment were proposed, including *L_A_*, *L*_50_ − *L*_90_ and *Sh*.

Based on previous research results, the mathematical relationship between “effective characteristics” and “satisfaction” is quantitatively studied in this paper. Using the multivariate linear regression modeling method, a subjective satisfaction evaluation model of the acoustic environment based on “effective characteristics” is built. After several residual optimizations, the regression accuracy of the model is more than 96%, which verifies the effectiveness of the built satisfaction evaluation model. Furthermore, representative typical open areas in Shanghai are selected to study the field application of the proposed evaluation model in real scenes. The result of the soundwalk experiment verifies that the prediction accuracy of the *Effective*
*Characteristics-Sat* evaluation model for soundwalk areas is more than 92% (obviously higher than the prediction accuracy of the traditional *L_A_-Sat* model), and the consistency of the satisfaction levels is more than 88%. The numerical simulation results show that the changes in the values of *Sh* and *L*_50_ − *L*_90_ have a significant impact on the satisfaction prediction of the proposed model when the *L_A_* is unchanged. That is, compared with the traditional subjective evaluation method based on a single sound pressure level, the proposed “effective characteristics” more comprehensively describe the quality level of the regional acoustic environment, and the satisfaction evaluation model based on “effective characteristics” has superior evaluation accuracy and regional applicability.

## Figures and Tables

**Figure 1 ijerph-19-09231-f001:**
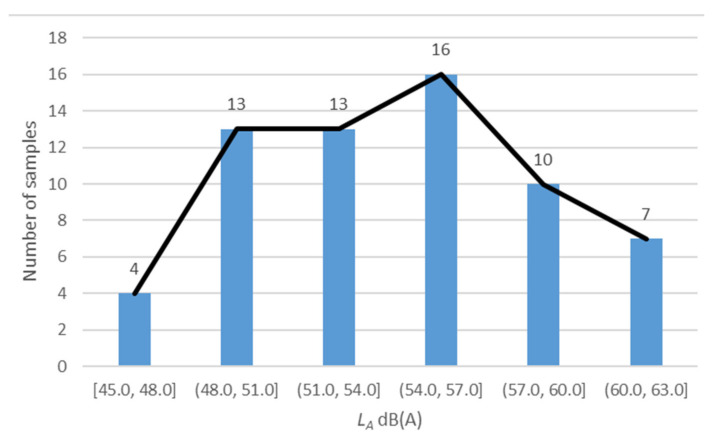
Statistical diagram of *L_A_* distribution of 63 acoustic environment samples.

**Figure 2 ijerph-19-09231-f002:**
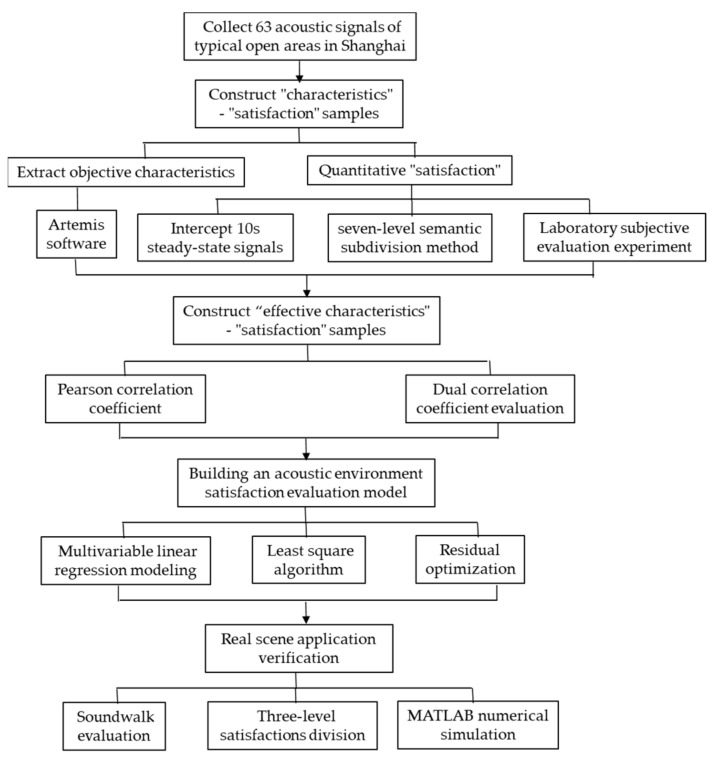
Flowchart of the methodology.

**Figure 3 ijerph-19-09231-f003:**
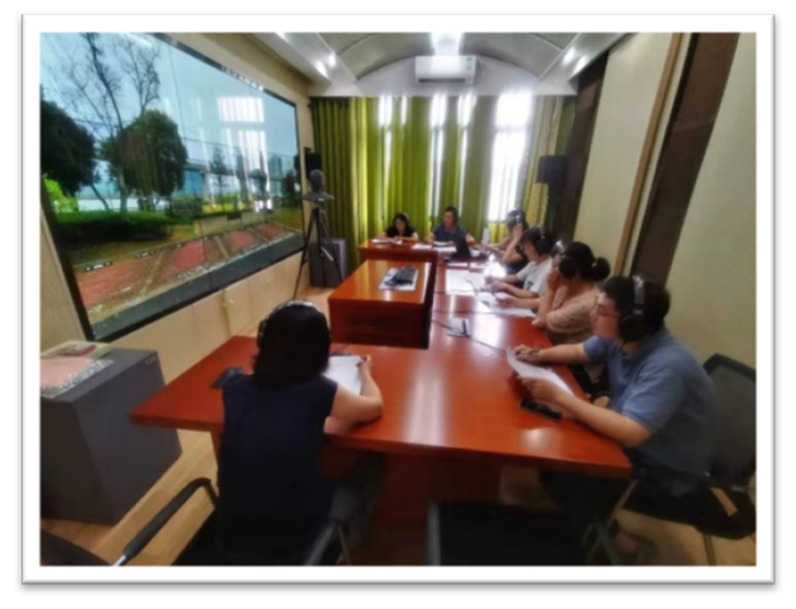
The scene of a subjective evaluation experiment in the laboratory.

**Figure 4 ijerph-19-09231-f004:**
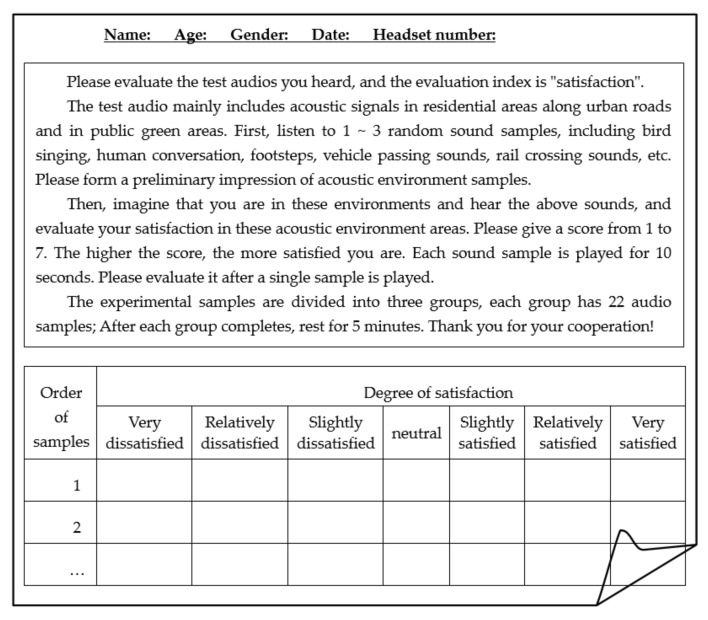
List of subjective evaluation experiments.

**Figure 5 ijerph-19-09231-f005:**
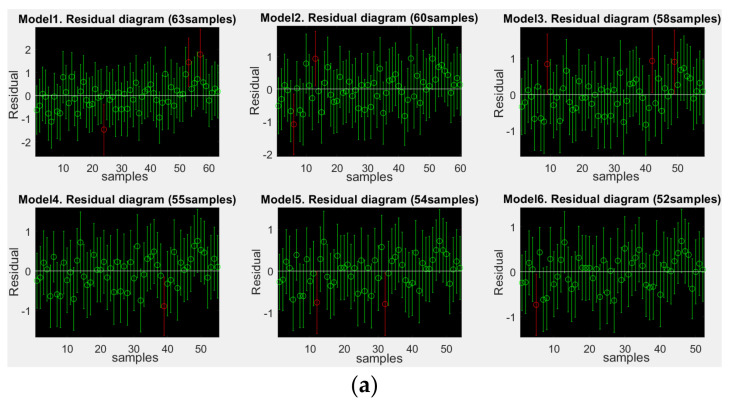
Time series residual diagrams of the initial regression model and the regression models with six optimizations. (**a**) Residual diagrams of initial regression and five times of optimization; (**b**) Residual diagram after six times of optimization.

**Figure 6 ijerph-19-09231-f006:**
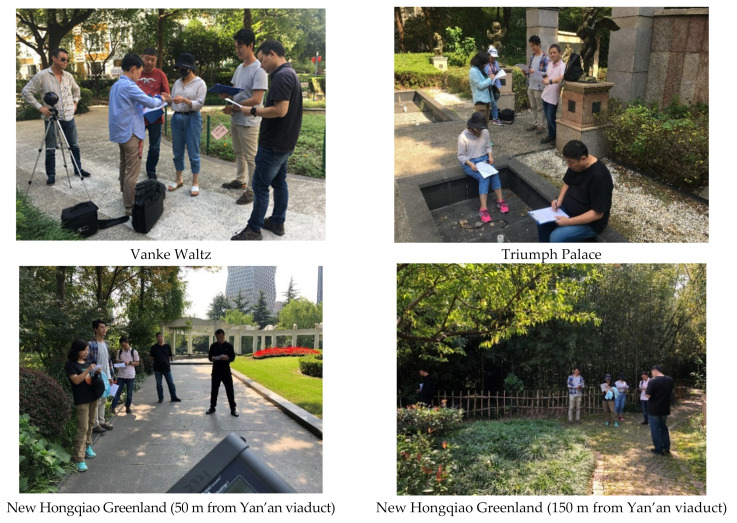
Scenes of the soundwalk experiment.

**Figure 7 ijerph-19-09231-f007:**
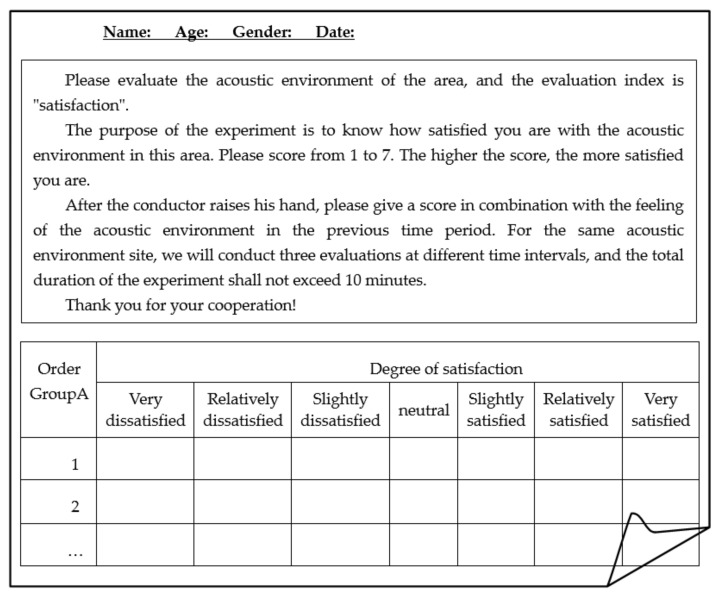
Subjective evaluation list of the field soundwalk experiment.

**Figure 8 ijerph-19-09231-f008:**
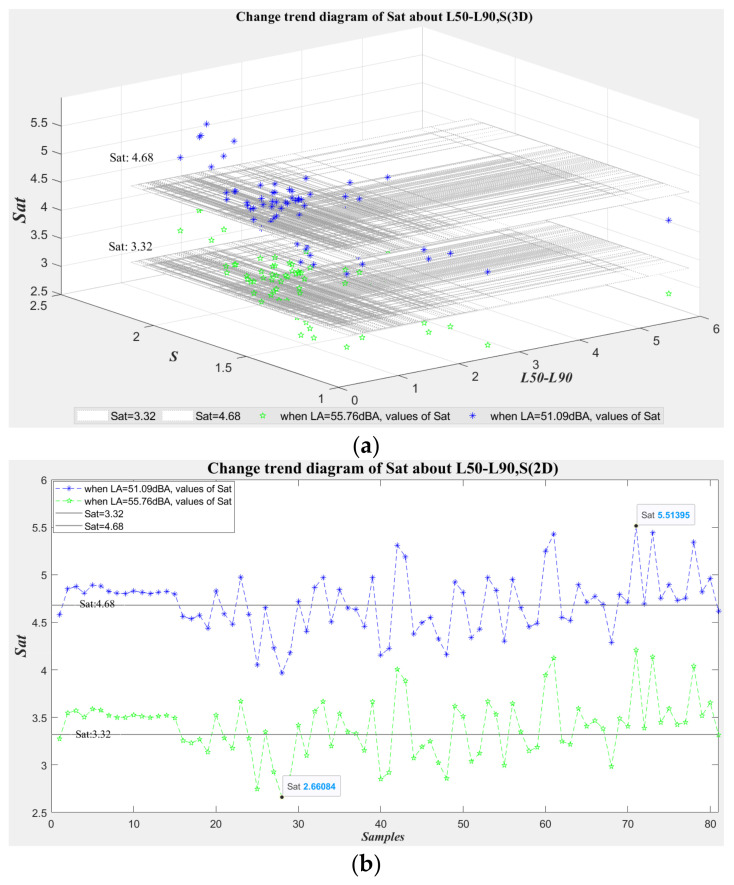
SAT_55.76dB(A)_, SAT_51.09dB(A)_ of 81 × 2 simulated samples (with different values of characteristics *L*_50_ − *L*_90_ and *Sh*) when *L_A_* is equal to 55.76 dB and 51.09 dB, respectively. (**a**) Change trends of SAT_55.76dB(A)_, SAT_51.09dB(A)_ on the characteristics *L*_50_ − *L*_90_ and *Sh* of 81 × 2 simulated sound samples (3D); (**b**) Change trend of SAT_55.76dB(A)_, SAT_51.09dB(A)_ on sample sequence index of 81 × 2 simulated sound samples (2D).

**Table 1 ijerph-19-09231-t001:** Correlation coefficients *R* (1 × 16) of 16 characteristic parameters with satisfaction (*Sat*).

*R* (1 × 16)	*L* _5_	*L* _10_	*L* _50_	*L* _90_	*L*_10_ − *L*_90_	*L*_50_ − *L*_90_	*L_A_*	*L_C_*	*L_C_* − *L_A_*	*D* _250_	*D* _315_	*D* _500_	*Flu*	*R*	*Lou*	*Sh*
** *Sat* **	−0.80	−0.85	−0.90	−0.87	0.02	−0.25	−0.88	−0.85	−0.13	−0.04	−0.03	0.06	0.15	−0.74	−0.87	0.33

**Table 2 ijerph-19-09231-t002:** Autocorrelation coefficients *R* (16 × 16) between 16 characteristic parameters.

*R* (16 × 16)	*L* _5_	*L* _10_	*L* _50_	*L* _90_	*L*_10_ − *L*_90_	*L* _50_ *−* *L* _90_	*L_A_*	*L_C_*	*L_C_* − *L_A_*	*D* _250_	*D* _315_	*D* _500_	*Flu*	*R*	*Lou*	*Sh*
** *L* _5_ **	1.00	0.99	0.90	0.83	0.34	0.37	0.96	0.71	−0.23	0.34	0.34	0.28	0.13	0.79	0.94	−0.07
** *L* _10_ **	0.99	1.00	0.95	0.89	0.26	0.37	0.98	0.77	−0.18	0.30	0.30	0.23	0.07	0.82	0.96	−0.14
** *L* _50_ **	0.90	0.95	1.00	0.97	−0.02	0.27	0.98	0.85	−0.03	0.17	0.17	0.10	−0.05	0.84	0.96	−0.27
** *L* _90_ **	0.83	0.89	0.97	1.00	−0.21	0.02	0.95	0.84	0.00	0.09	0.10	0.06	−0.08	0.84	0.92	−0.22
***L*_10_ − *L*_90_**	0.34	0.26	−0.02	−0.21	1.00	0.76	0.10	−0.13	−0.38	0.44	0.43	0.35	0.31	−0.03	0.11	0.15
***L*_50_ − *L*_90_**	0.37	0.37	0.27	0.02	0.76	1.00	0.28	0.16	−0.15	0.32	0.30	0.17	0.11	0.11	0.26	−0.23
** *L_A_* **	0.96	0.98	0.98	0.95	0.10	0.28	1.00	0.82	−0.11	0.24	0.23	0.18	0.02	0.85	0.97	−0.19
** *L_C_* **	0.71	0.77	0.85	0.84	−0.13	0.16	0.82	1.00	0.48	−0.33	−0.34	−0.40	−0.15	0.74	0.82	−0.34
***L_C_* − *L_A_***	−0.23	−0.18	−0.03	0.00	−0.38	−0.15	−0.11	0.48	1.00	−0.93	−0.94	−0.96	−0.29	−0.01	−0.07	−0.29
** *D* _250_ **	0.34	0.30	0.17	0.09	0.44	0.32	0.24	−0.33	−0.93	1.00	0.99	0.95	0.26	0.06	0.17	0.06
** *D* _315_ **	0.34	0.30	0.17	0.10	0.43	0.30	0.23	−0.34	−0.94	0.99	1.00	0.96	0.26	0.06	0.16	0.08
** *D* _500_ **	0.28	0.23	0.10	0.06	0.35	0.17	0.18	−0.40	−0.96	0.95	0.96	1.00	0.25	0.03	0.12	0.21
** *Flu* **	0.13	0.07	−0.05	−0.08	0.31	0.11	0.02	−0.15	−0.29	0.26	0.26	0.25	1.00	0.13	0.05	0.24
** *R* **	0.79	0.82	0.84	0.84	−0.03	0.11	0.85	0.74	−0.01	0.06	0.06	0.03	0.13	1.00	0.88	−0.02
** *Lou* **	0.94	0.96	0.96	0.92	0.11	0.26	0.97	0.82	−0.07	0.17	0.16	0.12	0.05	0.88	1.00	−0.07
** *Sh* **	−0.07	−0.14	−0.27	−0.22	0.15	−0.23	−0.19	−0.34	−0.29	0.06	0.08	0.21	0.24	−0.02	−0.07	1.00

**Table 3 ijerph-19-09231-t003:** Parameters of the multivariable linear regression model ($\alpha =0.05,p_{F}<0.0001$).

Regression Coefficient	Estimated Value	Confidence Interval
$\theta_{0}$	16.93	[15.391, 18.473]
$\theta_{1}$	−0.017	[−0.114, 0.080]
$\theta_{2}$	−0.28	[−0.303, −0.256]
$\theta_{3}$	1.3	[0.857, 1.738]

*R*^2^ = 0.9347, *F* = 224.4254, *p* < 0.0001, *S*^2^ = 0.1249.

**Table 4 ijerph-19-09231-t004:** *S*^2^ values, *RMSE* values and correlation coefficients $R_{\left(Sat,S\hat{a}t\right)}$ of 7 regression models (the three performance indicators of Model 7 are the best, which are bold).

Order	Number of Training Samples	*S* ^2^	*RMSE*	$R_{\left(Sat,S\hat{a}t\right)}$
Model 1	63	0.357	0.578	0.901
Model 2	60	0.240	0.473	0.934
Model 3	58	0.208	0.441	0.942
Model 4	55	0.169	0.396	0.954
Model 5	54	0.155	0.378	0.959
Model 6	52	0.135	0.352	0.965
Model 7	51	**0.125**	**0.339**	**0.967**

**Table 5 ijerph-19-09231-t005:** List of Regions, Points and Schemes of field soundwalk experiment.

Order	Region	Point	Scheme	Features
1	Vanke Waltz	50 m from Caobao Road	Conduct two field evaluations for 1 min and 5 min respectively	Residential areas along the large flow traffic artery of Caobao Road
2	Triumph Palace hinterland	170 m from rail transit line 3/4	Conduct field evaluations every 1 min for 10 min	Residential areas along rail transit line
3	New Hongqiao Greenland	50 m from Yan’an viaduct	Conduct three field evaluations for 1 min, 3 min and 5 min respectively	Greenland along the compound road of viaduct and ground
		150 m from Yan’an viaduct		

**Table 6 ijerph-19-09231-t006:** *MAPE* values of the *Effective Characteristics-Sat* model and *L_A_-Sat* model.

Number of Samples	18 Groups of Samples		14 Groups of Samples (Eliminate the Evaluation Results for 1-Min)	
**Subjective Evaluation Model**	*Effective Characteristics-Sat* Model	*L_A_*-*Sat* Model	*Effective Characteristics-Sat* Model	*L_A_*-*Sat* Model
** *MAPE* **	7.66%	10.06%	6.64%	9.84%
**1-*MAPE***	92.34%	89.95%	93.36%	90.16%

**Table 7 ijerph-19-09231-t007:** Comparison of prediction results and soundwalk evaluation results of acoustic environment satisfaction (“√” indicates that the satisfaction prediction value of the model is consistent with the evaluation value of the soundwalk, “×” indicates that the satisfaction prediction value of the model is inconsistent with the evaluation value of the soundwalk).

Point	Distance/m	Period/min	Effective Characteristics			Satisfactions				Consistency of Satisfaction Levels
			*L*_50_ − *L*_90_	*L_A_*	*Sh*	*Effective Characteristics-Sat* Model		Soundwalk		
						Prediction Value	Satisfaction Level	Soundwalk Value	Satisfaction Level	
Vanke Waltz	50	1	2.33	52.81	1.51	4.10	neutrality	4.50	neutrality	√
		5	2.40	53.07	1.72	4.30	neutrality	4.67	neutrality	√
Triumph Palace hinterland	170	1	0.93	52.35	1.72	4.52	neutrality	5.00	satisfaction	×
		2	1.18	55.40	1.67	3.60	neutrality	4.33	neutrality	√
		3	1.46	54.70	1.74	3.88	neutrality	4.00	neutrality	√
		4	1.47	54.50	1.73	3.93	neutrality	4.33	neutrality	√
		5	1.65	55.10	1.69	3.70	neutrality	3.67	neutrality	√
		6	1.35	54.70	1.67	3.80	neutrality	4.33	neutrality	√
		7	1.40	55.10	1.67	3.68	neutrality	3.67	neutrality	√
		8	1.35	54.95	1.69	3.75	neutrality	3.83	neutrality	√
		9	1.42	55.00	1.68	3.72	neutrality	3.67	neutrality	√
		10	1.44	54.95	1.67	3.72	neutrality	3.67	neutrality	√
New Hongqiao Greenland	50	1	0.60	56.45	1.67	3.32	dissatisfaction	3.17	dissatisfaction	√
		3	0.84	56.70	1.68	3.26	dissatisfaction	3.17	dissatisfaction	√
		5	0.92	56.95	1.66	3.16	dissatisfaction	3.17	dissatisfaction	√
	150	1	1.19	58.10	1.48	2.60	dissatisfaction	3.33	neutrality	×
		3	1.10	58.10	1.46	2.58	dissatisfaction	3.17	dissatisfaction	√
		5	1.14	58.40	1.49	2.53	dissatisfaction	3.00	dissatisfaction	√

## Data Availability

The datasets generated during the current study are available from the corresponding author on reasonable request.

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
