# Peer review of "Study on Subjective Evaluation of Acoustic Environment in Urban Open Space Based on “Effective Characteristics”"

_ijerph, 2022, doi:10.3390/ijerph19159231_

Round 1
Reviewer 1 Report
The manuscript is good, and clear chapters except conclusions that covers both discussion and conclusions. My suggestion is that these two chapters are separated and conclusions are the conclusions from this study. Statistical approach is good, and it proves the claims for the superiority. Perhaps the achieved and final equation of effective characteristics should be emphasized in the conclusion area also, now it can be forgotten inside the text, when effective characteristics is only mentioned as a general term.
Some minor things: line 124 verification, line 160 texts inside figures to English, page numbers do not work lines 211 and line 468 and line 481 shanghai?
Reviewer 2 Report
Dear authors,
This research is very interesting and novel. Congratulations!
Nevertheless, it needs major improvements regarding the English language. In several parts it was very difficult to understand. Nevertheless, I believe that this research is worthy of publication.
Please avoid large sentences as they are hard to follow. Also, avoid the repetition of specific words similar to “acoustic environment”. This repetition, and the large sentences vastly reduce the manuscript’s quality.
Additionally, it would be interesting to portray the differentiation between the concepts of “acoustic environment” and “soundscape” through your research.
Furthermore, only 33 references are used in this study. Please include the following (or similar) articles regarding the subjective and objective evaluation of the acoustic environment, in order to support the theoretical framework.
https://doi.org/10.3390/su13095284
https://doi.org/10.1121/10.0009794
https://doi.org/10.1016/j.buildenv.2022.109231
https://doi.org/10.1121/1.5102164
https://doi.org/10.1016/j.buildenv.2021.108456
https://doi.org/10.1016/j.buildenv.2021.107688
https://doi.org/10.3390/ijerph18063151
https://doi.org/10.1016/j.apacoust.2021.108447
https://doi.org/10.1016/j.scitotenv.2021.149869
https://www.who.int/europe/publications/i/item/9789289053563
I would strongly suggest revising your manuscript and make it more “easy to read”. Please rephrase your main goals and your main findings. Also highlight the differences of your research with other studies that achieved a combination of subjective psychoacoustic assessment metrics (also using evaluation Likert scales) and quantitative data similar to Leq.
Several suggestions are listed below.
Lines 9-12: The phrase is long and confusing. Please avoid using the phrase “acoustic environment” so many times.
Line 14: “at home”. The authors probably talk about Shanghai, China. Please maintain a global approach. It would be best: “In global literature, various subjective and objective acoustic characteristic parameters… have been introduced”.
Line 27: Furthermore
Lines 41-49: Please use references to support the claims made. WHO noise guidelines are suggested.
Line 86: Please revise. “Small scale research are difficult to be applied in order to evaluate the overall urban acoustic environment with….”.
Lines 96-100: Please revise. Very long and confusing sentence. Probably the authors could use the following: “Therefore, the extraction of the "effective characteristics" that are applicable to the overall acoustic environment evaluation have always been the key technical problem to be solved. The solution to this problem could lead towards the creation of a scientific and efficient subjective evaluation model for urban acoustic environments.”
Line 113: “Comparation” did the authors mean “comparison”?
Lines 128-129: In my opinion numerous sub-sections should be avoided.
Line 130: It is not clear what class 2 sound functional areas are. Please provide more information and literature. “Ministry of environmental protection. Acoustic environmental quality standards[R]. 2008-10-01”.
Lines 137-147: This is one sentence. Please revise and shorten
Line 157: Use the word “neutral” instead of general
Line 160: Please translate figure 1
Line 197: Construction of "Effective Characteristics" - "Satisfaction" Sample set
Line 204: Please rephrase in order to be clearer “is the i-th”. It would be better to give more space to the formulas presented in order to be more clear, similar to line 224.
Line 215: “Building an and acoustic environment satisfaction….”
Line 250-252: The use of the word “take” is grammatically incorrect
Line 309: Please elaborate on what is a sound scene (maybe this is helpful: https://link.springer.com/chapter/10.1007/978-3-319-63450-0_1) and please explain and reference the International Standard of sound scene.
Lines 324-326: Phrase seems out of place
Line 341: They evaluated the degree of satisfaction
Line 342: Commander = soundwalk conductor or moderator
Line 345: Please explain “typical” in table 6.
Line 389: As shown
Lines 393-395: Please revise the sentence
Lines 399-404: The sentence is too long and confusing. Please revise
Lines 416-421: Long and confusing sentence. Please shorten and revise
Lines 431-434: “In combination with the 81 characteristic values of L50-L90 and S collected in the previous experiments, two groups of simulation experiment sample sets (each group containing 81 433 samples) were constructed using MATLAB”.
Line 446: “Change little” to “Slightly differ”
Line 452: “as it can be seen in Figure 5”
Lines 452-456: Please revise. It is very hard to understand
Reviewer 3 Report
Dear authors, thank you for sending this manuscript about the subjective evaluation of the environmental soundscape. This paper requires minor changes as per the following:
- Please specify if you are referring to sound pressure level or sound power level. Sound level is not clear enough.
- Figure 1: please change the axes title to English; no other languages are acceptable. The title of the graph shall not appear since there is a caption. Please improve the quality of this graph, may be changing the its format instead of bars.
Table 1: please delete all the symbol that are visible in blank cells, other than after the text. all the "enter symbols" shall be disappear as well as the other ones.
Table 2: as per above comments
Thank you
Reviewer 4 Report
The article “Study on subjective evaluation of acoustic environment in urban open space based on “effective characteristics”” investigates the subjective evaluation of the overall acoustic environment, Starting from the study of the correlations of some acoustic environment objective characteristic parameters and the subjective "satisfaction" parameter. The results obtained define three "effective characteristic" parameters for subjective evaluation of acoustic environment, i.e. LA, L50-L90 and Sharpness (S).
Some additional remarks:
- - Sentence at lines 86-77 “The researches with relatively small scale, so it is difficult to be 86 applied to evaluate the urban overall acoustic environment with complex and mixed 87 sound sources.” Should be reorganized.
- - It is suggested to pay attention to mistakes related to capital letters (e.g.: lines 124, 481, etc.);
- - Table 1 and 7 are not very clear, but need a higher graphic resolution;
- - The title and the names of the axes in Figure 1 should be in English;
- - More information about Artemis software, the subjective evaluation laboratory and experimental instruments would be appreciated;
- - It is suggested to rename section 2 with “Materials and methods”;
- - Section 2 should be organized in other two subparagraphs, which should be placed at the beginning, named: 2.1 Case study and 2.2 Methodology. Then the current “2.1. Construct Samples of "Effective characteristics" - "Satisfaction"” will be the new 2.3 paragraph, and so on;
- - In the new subparagraph 2.1 Case study, you should move the current lines 130- 135. Moreover, a map representing the areas of measurements would be appreciated;
- - In the new subparagraph 2.2 Methodology, you should move the current lines 101- 119, opportunely reorganized. Moreover, a flowchart scheme of the methodology, integrated by all the tools and methods that will be used at every steps, would be appreciated;
- - It is suggested to put figures and tables in the right position and order according to their reference in the text.
